# What Role Does COA6 Play in Cytochrome *C* Oxidase Biogenesis: A Metallochaperone or Thiol Oxidoreductase, or Both?

**DOI:** 10.3390/ijms21196983

**Published:** 2020-09-23

**Authors:** Shadi Maghool, Michael T. Ryan, Megan J. Maher

**Affiliations:** 1School of Chemistry and The Bio21 Molecular Science and Biotechnology Institute, The University of Melbourne, Parkville 3010, Australia; 2Department of Biochemistry and Molecular Biology, Biomedicine Discovery Institute, Monash University, Clayton 3800, Australia; michael.ryan@monash.edu; 3Department of Biochemistry and Genetics, La Trobe Institute for Molecular Science, La Trobe University, Melbourne 3086, Australia

**Keywords:** mitochondria, cytochrome *c* oxidase, assembly factor, COA6, copper, structure

## Abstract

Complex IV (cytochrome *c* oxidase; COX) is the terminal complex of the mitochondrial electron transport chain. Copper is essential for COX assembly, activity, and stability, and is incorporated into the dinuclear Cu_A_ and mononuclear Cu_B_ sites. Multiple assembly factors play roles in the biogenesis of these sites within COX and the failure of this intricate process, such as through mutations to these factors, disrupts COX assembly and activity. Various studies over the last ten years have revealed that the assembly factor COA6, a small intermembrane space-located protein with a twin CX_9_C motif, plays a role in the biogenesis of the Cu_A_ site. However, how COA6 and its copper binding properties contribute to the assembly of this site has been a controversial area of research. In this review, we summarize our current understanding of the molecular mechanisms by which COA6 participates in COX biogenesis.

## 1. Introduction

Mitochondria are double membrane-bound organelles required for life in most eukaryotes, including humans. Mitochondria possess a unique architecture composed of compartments that include the outer membrane (OM), intermembrane space (IMS), inner membrane (IM), the cristae, and the matrix [1,2,3]. The mitochondrial oxidative phosphorylation (OXPHOS) system, located in the mitochondrial IM, is composed of five multi-subunit protein complexes including NADH:ubiquinone oxidoreductase (complex I), succinate:ubiquinone oxidoreductase (complex II), ubiquinol:cytochrome *c* oxidoreductase (complex III), cytochrome *c* oxidase (complex IV; COX), and the F_o_F_1_ ATP synthase (complex V) [4,5,6,7].

COX, a multi-subunit oxidoreductase of dual genetic origin, is the terminal electron acceptor in the OXPHOS system. COX catalyzes the reduction of molecular oxygen to water and therefore contributes to the generation of the proton motive force that is utilized by complex V for ATP synthesis [6,8]. COX is composed of 14 subunits where the catalytic core, conserved from α-proteobacteria to humans, contains 3 subunits (COX1, COX2, and COX3) encoded by mitochondrial DNA. The remaining 11 subunits are encoded by nuclear DNA and are imported into mitochondria following translation in the cytosol [6,7,9].

The elucidation of the crystal structure of COX from bovine heart mitochondria (PDB 1OCC) [5] was a major breakthrough in the field of mitochondrial research (Figure 1a). The structure showed that the COX1, COX2, and COX3 subunits are all integral membrane proteins. COX1 and COX3 are predominantly hydrophobic, whereas COX2 includes a β-barrel extra-membrane domain, which extends into the IMS to interact with cytochrome *c*. This domain is connected to two transmembrane α-helices which create an interface between the COX2 and COX1 subunits. The COX1 and COX2 subunits bind the catalytic metal centers of the enzyme. COX1 contains a low spin heme *a*, a mononuclear Cu_B_ site, and a high spin heme *a*_3_ cofactor, while COX2 contains the dinuclear Cu_A_ site [8,10]. Cytochrome *c* is oxidized at the dinuclear Cu_A_ site and electrons are transferred to the heme *a* in COX1, which in turn transfers the electrons to the heme *a*_3_ and mononuclear Cu_B_ site. This is the site where O_2_ is reduced to water [6,8].

Various COX assembly factors are required for subunit maturation, cofactor incorporation, and for the stabilization of assemblies (or modules) that are intermediates of COX biogenesis. A total of 36 human COX assembly factors have been identified to date [8,11], but this list continues to be extended. Earlier studies proposed that COX assembly occurs via a linear process where the various subunits and cofactors are added in a sequential manner [11,12]. However, the most recent analyses suggest that COX assembly occurs as a combination of modular and linear processes, in which the biogeneses of the catalytic core subunits are relatively independent processes. In this model, specific chaperones function in the assembly of each subunit and modules containing different subunits associate in an ordered manner [7,13,14,15]. Succinctly, human COX assembly initiates with the independent formation of core modules that include: (1) a COX1-containing module with assembly factors COX14, COA3, COA1, and SURF1; (2) a COX2-containing module with subunits COX5B, COX6C, COX7B, COX7C, and COX8 and (3) a COX3-containing module with subunits COX6A, COX6B, COX7A, and NDUFA4. Finally, the formation of the fully assembled COX occurs by engagement of COX1 with the COX4 and COX5a subunits prior to assembly with the COX2 and COX3 modules and remaining subunits [7,14,15].

## 2. The COX Metal Centers

### 2.1. COX1 Binds Heme and Copper

Hemylation of the COX1 core subunit is crucial for oxidase activity, with COX the only mitochondrial enzyme that requires heme *a* as a cofactor. COX10 (heme *o* synthase) and COX15 (heme *a* synthase) are integral mitochondrial IM enzymes that catalyze heme *a* biosynthesis from heme *b* [16,17]. However, the precise mechanism of subsequent insertion of heme *a* into COX1 remains unclear.

The mononuclear Cu_B_ site, which is located 4.7 Å from the heme *a*_3_ co-factor, binds a single copper atom via coordination with three histidine ligands (Figure 1b) [18]. The metallochaperone COX11 is the major chaperone for Cu(I) delivery to the Cu_B_ site. COX11 is composed of a single transmembrane α-helix connected by a flexible linker (15 residues) to a soluble IMS-located C-terminal domain, which includes a Cys-xx-Cys motif for Cu(I) binding [18]. The redox states of the Cys residues in the Cys-xx-Cys motif cycle between oxidized (a disulfide bond) and reduced forms (free Cys residues for Cu(I) binding), catalyzed by COX19, which also contains a CX_9_C motif [19]. During copper transfer, the COX11 dimer has been proposed to position above the membrane surface with its Cu(I) sites facing the membrane, which facilitates the transfer of Cu(I) to the Cu_B_ site of the COX1 subunit [20,21].

### 2.2. COX2 Binds the Dinuclear Cu_A_ Cite

The binuclear Cu_A_ site of COX2 is situated in its IMS-located α-barrel domain, which is close to the COX1–COX2 interface. The two copper atoms are bridged by two cysteine residues (Cys196 and Cys200). One copper atom is coordinated by the imidazole group of residue His161 and the thioether group of Met207, while the other copper atom is coordinated by residue His204 and the carbonyl group of Glu198 (Figure 1c) [8,22]. Metalation of COX2 occurs in the mitochondrial IMS and several assembly factors, which bind copper and have redox activities, including COX17, SCO1, SCO2, COA6, and the newly identified COX16 protein, are required for the biogenesis of the Cu_A_ site [18,23,24,25]. Insertion of copper into COX2 requires the reduction of a disulfide bond between COX2 residues Cys196 and Cys200, in addition to copper transfer from metallochaperone proteins.

Impaired biogenesis of COX is a cause of human mitochondrial disease [26], where mutations in a number of assembly factors have been identified in patients. For example, patients with mutations in the SCO1, SCO2, and COA6 proteins suffer from hypertrophic cardiomyopathy owing to a defect in COX2 biogenesis which results in COX deficiency [18,23,24,25,27].

## 3. Biogenesis of the Cu_A_ Site of COX2

### 3.1. COX17 Distributes Cu(I) in the Mitochondrial IMS

The Cu(I)-metallochaperone COX17 is a soluble IMS protein (62 residues) with a twin CX_9_C motif sequence motif [28]. The NMR structure of COX17 has been determined (PDB 2RNB) [28] and shown to adopt a coiled-coil–helix–coiled-coil–helix (CHCH) fold. COX17 possesses six cysteine residues, with three disulfide bonds between the three cysteine pairs in the fully oxidized state (COX17_3S-S_). The cytosolic, fully reduced COX17 is unstructured. Once COX17 is imported into the IMS, it is partially oxidized by Mia40 to COX17_2S-S_, and binds a single Cu(I) atom between the surface-exposed pair of Cys residues (Cys22 and Cys23) [23,28]. COX17 receives copper from the mitochondrial matrix copper pool [29], where copper is bound by an anionic fluorescent molecule (also known as copper ligand (CuL)) [30]. It has been reported that CuL is also found in the cytoplasm and may provide a mechanism for copper translocation into the mitochondria, the mechanism for which remains unknown [29]. In yeast (*Saccharomyces cerevisiae*), there are two copper transporters, the mitochondrial phosphate carrier protein Pic2 [31] and the mitochondrial inner membrane iron transporter Mrs3 [32], which act in copper transfer from the IMS to the matrix for storage. In humans, the transporter responsible for this process is SLC25A3, a member of the mitochondrial phosphate carrier family of proteins (Pic) [33,34]. Misregulation of copper homeostasis has been shown to have a broad effect on mitochondrial function, where it has been proposed to damage mitochondrial [4Fe-4S] protein maturation [35]. Despite recent progress in elucidating a detailed mechanism for copper delivery to COX, the mechanism by which COX17 receives copper within the IMS remains to be defined. Nevertheless, once COX17 is loaded with copper, the metal is distributed to both the Cu_A_ site in COX2 and the Cu_B_ site in COX1 [23,36,37].

### 3.2. The SCO Proteins Facilitate Metalation of the COX2-Cu_A_ Site

Both the SCO1 and SCO2 proteins play crucial roles in metalation of the COX2-Cu_A_ site and mutations in either protein lead to a COX assembly defect and mitochondrial disease [38,39,40]. Structurally, both SCO proteins are composed of a single transmembrane α-helix in addition to a soluble IMS-located globular domain. NMR structures of the soluble domains of SCO1 (PDB 2GQM) [41] and SCO2 (PDB 2RLI) [42] have been determined in the presence of Cu(I). In both proteins, a CX_3_C motif and a conserved histidine residue bind either Cu(I) or Cu(II), with the Cu(II) binding properties of the SCO proteins characterized by EPR spectroscopy [43]. The SCO1 and SCO2 cysteine residues within the CX_3_C motifs form disulfide bonds with redox potentials of −277 mV [44] and less than −300 mV [42], respectively, with the reduced thiols coordinating copper. The highly negative redox potentials of the SCO proteins correlate with the ability of these proteins to reduce the disulfide bond between residues Cys200 and Cys204 within the Cu_A_ site of COX2 to facilitate copper binding. However, a recent study that employed NMR spectroscopy with 4- acetoamido-4′-maleimidylstilbene-2,2′-disulfonic acid (AMS) electrophoretic mobility shift assays [45] revealed that Cu(I)-bound SCO2 acts as a thiol reductase to reduce the Cys residues of a chimeric COX II*_S-S_ protein, constructed by inserting the three loops of the human COX2 into the stable scaffold of the *Thermus thermophilus* COX2. In contrast, the metallochaperone SCO1 was not able to reduce the chimeric COX II*_S-S_ and instead was shown to transfer copper to the reduced form of the protein [45]. This reinforced earlier proposals [23] of unique, non-redundant functions for the human SCO1 and SCO2 proteins, being copper transfer and cysteine disulfide oxidoreductase activities, respectively. Several studies have shown that an additional assembly factor, COA6, interacts with the SCO proteins and newly synthesized COX2 for the assembly of COX [25,46,47].

## 4. COA6 is a COX Assembly Factor

### 4.1. COA6 is an IMS Protein that Functions in COX Assembly

COA6, also known as C1orf31, was first identified as a potential COX assembly factor by an iterative orthology prediction method [48]. Subsequently, a proteomic survey of mitochondrial IMS proteins in *Saccharomyces cerevisiae* revealed that Coa6 is a soluble IMS located protein with CX_9_C–CX_10_C motif [49]. The protein Ymr244c-a (the C1orf31 yeast homologue) was termed cytochrome *c* oxidase assembly factor 6 (Coa6) as it was established that its absence in yeast caused specific COX assembly defects [49]. Similar to its yeast homolog, studies using human cell lines showed that human COA6 is essential for COX biogenesis and that its loss leads to a severe COX assembly defect [25,46]. Immunoprecipitation studies using yeast and human cell lines showed that COA6 interacts with newly translated COX2 and is required for its maturation [25,46,50].

A targeted next-generation sequencing study identified two pathogenic mutations of the COA6 protein, W59C and E87*, in a patient suffering from hypertrophic obstructive cardiomyopathy with a combined mitochondrial complex I and COX deficiency in the heart tissue and no defect in the fibroblasts [51]. These mutations are found in conserved residues within the CX_9_C–CX_10_C motif. Blue native polyacrylamide gel electrophoresis (BN-PAGE) and western blotting analyses of isolated mitochondria from a heart biopsy and the patient skin fibroblasts revealed the absence and presence of COX, respectively in the heart and fibroblast mitochondria. This suggested that COA6 mutations result in a tissue-specific COX defect [25]. The missense W59C and nonsense E87* mutations were mimicked in yeast, which established that the variants could not rescue the respiratory growth defect of yeast Coa6^KO^ cells [50]. However, transient overexpression of the pathogenic mutation ^W59C^COA6 could partially restore COX assembly in human COA6^KO^ cells [25,52]. In a separate study, a patient with W66R mutation in COA6 was identified with neonatal hypertrophic cardiomyopathy, muscular hypotonia, and lactic acidosis and a clear COX defect in fibroblasts [53]. Overexpression of the W66R variant in patient fibroblasts did not rescue COX activity [52].

### 4.2. COA6 Interacts with Other COX Assembly Factors Associated with Biogenesis of the Cu_A_ Site

Our previous research [25] and that by others [46,47] have sought to define interacting partner proteins of COA6 in order to define its role in COX assembly. It has been suggested that COA6 forms a complex with the SCO proteins where copper transfer from SCO1 to Cu_A_ site is facilitated by SCO2 and COA6 [45,46,47,50]. However, the identities of the interactors are controversial, reflecting the general uncertainty regarding the precise order of interactions and events that facilitate the biogenesis of the COX Cu_A_ site. Various reports have proposed that that COA6 specifically interacts with SCO1 [25], or SCO2 [46], or both SCO1 and SCO2 [47] and a recent study asserted that in the presence of both proteins, COA6 preferentially interacts with SCO1 over SCO2 [52]. Besides the SCO proteins, COA6 has also been shown to associate with other COX assembly factors such as COX16, which is required for COX2 biogenesis and maturation [27] and COX18 and COX20 [54]. The COX18 and COX20 proteins are required for membrane insertion and translocation of the C-terminus of COX2 across the mitochondrial IM [22,54]. In summary, COA6 interacts with a host of COX assembly factors in addition to the COX2 subunit of COX. Whether these interactions indicate a role for COA6 in copper transfer, thiol redox activity, and/or involve the formation of a COX assembly complex that includes COA6 and one or more of the SCO1, SCO2, COX16, COX18, or COX20 proteins remains unclear.

### 4.3. COA6 and Copper

A number of recent studies point to a role for COA6 in copper delivery to COX and therefore assembly of the Cu_A_ site. Respiratory growth defects observed for Coa6^KO^ and Cox17^KO^ yeast were shown to be completely rescued by exogenous copper supplementation, but not by zinc, magnesium, or cobalt [50]. Accordingly, treatment of COA6-deficient patient fibroblasts with CuCl_2_ revealed partial rescue of the observed COX levels [53]. Finally, the anticancer drug elesclomol (ES) was proposed to restore COX function by increasing mitochondrial copper content in patient cells with mutations in the COA6 and SCO2 proteins [47,55].

A number of studies have investigated the Cu-binding properties of COA6, in vitro. We showed that recombinant COA6 binds Cu(I) with high affinity (K_D_ ~10^−17^ M ) [25] and that mutagenesis of residues Cys58 and Cys90 (to Ser) eliminates Cu(I) binding [56]. In yeast, Coa6 was shown to interact with Cox2 in a copper–chaperone-dependent manner, as the interaction between Sco1/2 and Coa6 was amplified in the absence of Cox2. This suggested that Coa6 accumulates in a complex with the Sco1 and Sco2 metallochaperones in the absence of its substrate. In addition, atomic absorption spectroscopy revealed that overexpressed forms of Coa6, Sco1, and Sco2 purified from *E. coli* were bound to Cu and at similar stoichiometries [46]. While this observation was supported by other studies [25,46,47], it was recently concluded that Coa6 does not bind copper in yeast under physiological conditions since mitochondria from both wild-type and the Coa6^KO^ yeast cells contained similar copper contents [52]. However, why these data did not reflect a loss of total copper due to the absence of COX biogenesis in these mitochondria was not addressed.

### 4.4. The Thiol Oxidoreductase Activity of COA6

Mitochondrial IMS-located proteins with twin CX_9_C motifs are commonly redox active through the cycling of cysteine pairs between reduced (SH) and oxidized (S-S) forms [57,58]. In addition, high-affinity Cu(I)-binding metallochaperones possess Cys-x_n_-Cys motifs where Cu(I) binds via coordination with the thiol groups (SH) of the Cys residues when in the reduced state [59]. The redox potentials of the Cys pairs therefore affect the copper binding and delivery activities of these metallochaperones and should be considered, along with relative protein–metal binding affinities, when attempting to predict copper transfer from one protein partner to another [60,61,62]. Previously, the redox potentials of COX17, SCO1, COX2, and SCO2 have been determined as −198 mV, −277 mV, −290 mV, and less than −300 mV, respectively [42,44,45,57]. The redox potential of COA6 has been reported in two separate, studies as −349 ± 1 mV [56] and −330 mV [52], which is lower than the other proteins (COX17, SCO1, COX2, and SCO2) involved in copper transfer to COX. In agreement with these values, COA6 has been shown to reduce the critical disulfide bonds in COX2 and the SCO proteins [52,63]. This indicates that COA6 can act as a thiol reductase, similar to the recently defined function of SCO2. However, SCO2 requires Cu-binding for its thiol reductase activity, where reduction of the cysteine ligands of the Cu_A_ site is facilitated by redox cycling of the copper atom bound to SCO2 and not the Cys disulfide [45]. Although the thiol reductase activity of COA6 has not yet been determined in the presence of copper, in mitochondria, COA6 exists in a partially reduced state, which suggests that just one of the two disulfides is reduced and this could correlate with copper binding [25].

### 4.5. The Molecular Structure of COA6

The structure of COA6 has been described by X-ray crystallography (PDB 6PCE) [56] and NMR (PDB 6NL3) [52]. Consistent with previous size exclusion chromatography data [25], the crystal structure, determined to 1.65 Å resolution, shows a dimeric COA6 assembly with a tight dimer interface, which is modulated by both electrostatic and hydrophobic interactions. Each COA6 protomer shows a coiled coil–helix–coiled coil–helix (CHCH) fold [64,65], with the two N-terminal helices (α1 and α2, Figure 2) tethered at each end by disulfide bonds, between cysteine pairs Cys58-Cys90 and Cys68-Cys79. The third helix (α3, Figure 1) mediates the dimer interface.

The solution structure, determined by NMR (Figure 2b; [52]) is significantly different from that determined by crystallography (the root mean squared deviation for 69 common Cα positions on superposition of the two structures is ~12 Å; Figure 2). When the N-terminal helix (α1, Figure 2) of the monomeric crystal and solution structures are superposed, the topologies of the two structures are inconsistent. Specifically, helices α1 and α2 in the NMR structure are significantly shorter than those in the crystal structure (11 and 10 residues compared with 21 and 16 residues, respectively) and are linked by longer loops. Consequently, where the disulfide bonded cysteine pairs Cys58-Cys90 and Cys68-Cys79 are positioned on helices α1 and α2 in the crystal structure, in the solution structure, only Cys58 and Cys79 are located similarly, with the remaining Cys residues on loops between the helices. Perhaps most significantly, where helix α3 in the crystal structure is continuous (18 residues long), in the solution structure, the helix is ‘broken’ (α3 and α3*, Figure 2b) and interacts with the opposite ‘face’ of the α1/α2 helical pair. For example, in the crystal structure, there are salt bridges between residues Arg83 (α2) and Asp99 (α3) and residues Asp70 (α1) and Arg102 (α3), which presumably stabilize the relative orientations of these secondary structural elements (Figure 2a, (ii)). In the NMR structure these residues are all surface exposed, on opposite sides of the molecule and separated by ~25 Å (Figure 2b, (ii)). Finally, although the crystal structure as determined by X-ray crystallography shows significant structural similarity to the COX6B subunit of COX, which also shows a CHCH fold [56], the solution structure does not.

Unfortunately, since no experimental data were deposited or included with the description of the solution structure, it is difficult to ascertain whether these observed differences are due to the conditions under which the structures were determined, conformational changes that occur between the solid and solution states, or other unknown factors. With these differences in mind, the modelling of the interaction of COA6 with partner proteins such as SCO1 which accompanied the description of the solution structure [52], and which formed the basis of the associated mechanistic proposal, would most likely yield different outcomes for the COA6 crystal versus solution structures. This reinforces the uncertainty regarding the interactions of COA6 with other assembly factors (discussed above) and therefore the precise, mechanistic role of COA6 in COX biogenesis.

### 4.6. The Structural Consequences of the COA6 Pathogenic Mutations

As detailed above, COA6 pathogenic mutations, linked to defects in COX assembly have been identified in patients with mitochondrial disease. These include the W59C, E87*, and W66R mutations in patients that suffered from hypertrophic obstructive cardiomyopathy, muscular hypotonia, and lactic acidosis [51,53] (Figure 3a). In conjunction with the description of the crystal structure of COA6, the structure of the W59C mutant protein was also described [56]. The W59C mutation was shown to accompany a significant change in the quaternary structure of the protein, from a noncovalent dimer for ^WT^COA6 to a dimer of dimers (a noncovalent dimer of disulfide bridged dimers) for ^W59C^COA6 (Figure 3b). The oligomerization of the ^W59C^COA6 protein was proposed to disrupt critical interactions with partner proteins and correlated with the observation that the ^W59C^COA6 forms aggregates in the mitochondrial IMS [25] or is mistargeted to the mitochondrial matrix [46]. These impacts of the mutation were proposed to account for the pathogenic effects of this variant.

For the E87* mutation, both the COA6 crystal and solution structures show that residue E87 is located on helix α2, indicating that this mutation would result in a truncated form of the protein (lacking helix α3) that would presumably be non-functional in COX biogenesis. Finally, for the ^W66R^COA6 mutation, the COA6 crystal structure shows that the W66 side chain sits between helices α1 and α2 and stacks between the sidechains of residues Arg62 and Arg102. The introduction of this mutation would potentially disrupt the inter-helical packing through repulsion between the introduced Arg66 side chain and its neighbors. In fact, recently, we examined the structure and stability of the recombinant ^W66R^COA6 protein by 1D NMR spectroscopy and differential scanning fluorimetry (DSF; unpublished data), which showed the ^W66R^COA6 protein to have a significantly lower melting temperature, indicating lower stability. This agrees with a recent study showing that the overexpression of this variant in patient fibroblasts could not rescue COX activity [52].

## 5. Conclusions

Despite the importance of COX assembly in health and mitochondrial disease, we have only a limited understanding of the molecular basis of its biogenesis, due to a lack of knowledge about structures and precise functions of the individual COX assembly factors. Assembly of the COX2-Cu_A_ site requires the action of assembly factors that facilitate both thiol oxidoreductase and copper trafficking processes. There exists a general consensus, built from a number of comprehensive studies, that COX17 acts in copper delivery to the SCO1 and SCO2 proteins and that their proposed non-redundant activities (SCO1 in copper trafficking and SCO2 in disulfide reduction) lead to copper insertion into the Cu_A_ site of COX. However, the roles of recently identified assembly factors such as COA6 and COX16, which have been shown experimentally and in patient samples to be crucial for COX biogenesis and activity, remain undefined. In addition, whether all assembly factors that contribute to COX biogenesis have been identified is an open question.

Certainly, the recent intense interest in the characterization of the structure and function of COA6 has yielded exciting insights. A number of studies have indicated that COA6 binds copper, although whether that property is directly linked to function remains controversial. Equally, the measured redox potential of the Cys disulfide in COA6 indicates that COA6 would be capable of catalyzing the reduction of disulfide bonds within assembly factor proteins and COX2, which is a well-recognized requirement for copper transfer. Whether this thiol oxidoreductase activity of COA6 requires or is augmented by copper binding is a possibility yet to be investigated, as is the possible direct transfer of copper to or from COA6 and other assembly factors and COX2. Future investigations should focus on these functional and mechanistic aspects. The ability to probe the metal contents, interactions, and the redox status of these proteins in real time in active mitochondria would resolve much of the current uncertainty.

## Figures and Tables

**Figure 1 ijms-21-06983-f001:**
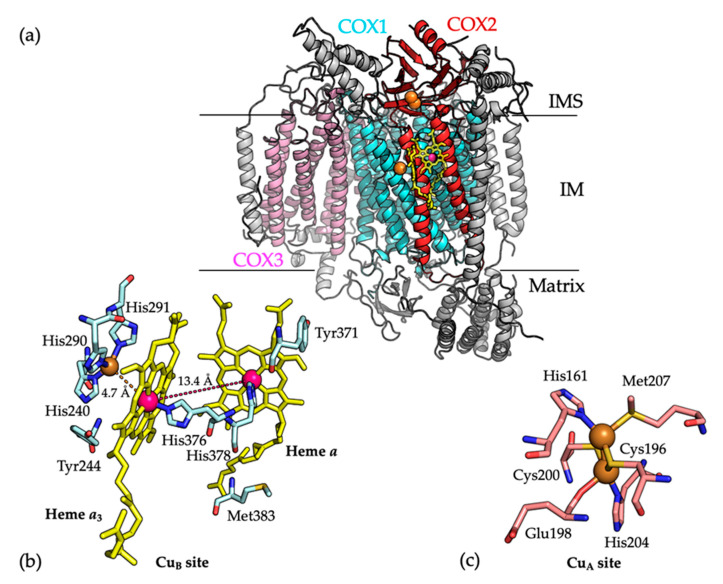
(**a**) The overall structure of the monomeric bovine COX (PDB 1OCC) [5]. All secondary structures are shown as cartoons. COX1, COX2, and COX3 subunits are colored in cyan, red and pink, respectively. Copper ions are shown as orange spheres and heme *a* and heme *a*_3_ are shown as yellow sticks. Other remaining subunits are colored in gray for clarity. (**b**) The mononuclear Cu_B_ site, which is located in close proximity to the heme *a*_3_ (4.7 Å), binds one copper ion via coordination with residues His240, His290, and His291. Residues located (labeled) at the Cu_B_ site are shown as cyan sticks. The copper (Cu) and iron (Fe) atoms are shown as orange and pink spheres, respectively. Heme *a* and heme *a*_3_ are shown as yellow sticks, where the distance between their Fe atoms is 13.4 Å which is shown as a dashed line. Carbon, oxygen, nitrogen and sulfur atoms are colored cyan, red, blue and yellow, respectively. (**c**) Residues located at the Cu_A_ site (labeled) are shown as sticks. Copper ions are shown as orange spheres. A cluster of two copper atoms, bridged by Cys196 and Cys200 residues constitutes the Cu_A_ site. One copper ion coordinated by the imidazole group of His161 and the thioether group of Met207 while the other copper ion is coordinated by His204 and the carbonyl group of Glu198. Carbon, oxygen, nitrogen and sulfur atoms are colored cyan, red, blue and yellow, respectively.

**Figure 2 ijms-21-06983-f002:**
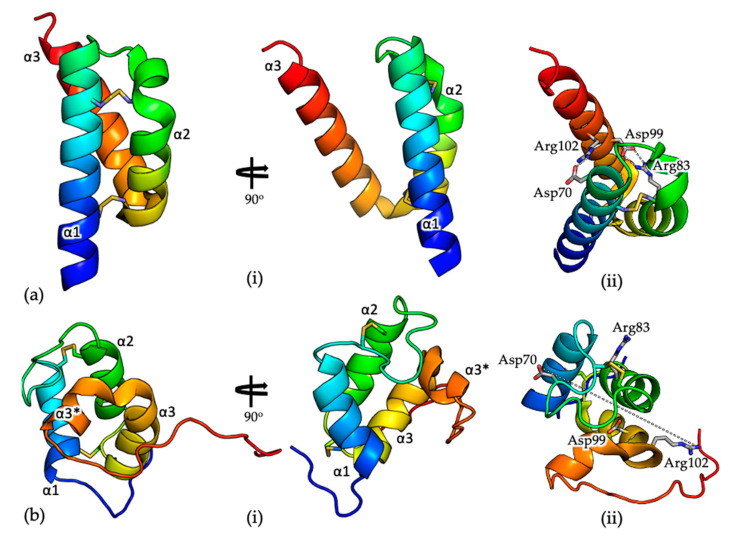
(**a**) Cartoon representation of the crystal structure of COA6 (PDB 6PCE) [56]. (i) The structures are colored from blue at the N-terminus to red at the C-terminus. Cysteine residues are shown as yellow sticks; (ii) Salt bridges between residues Arg83 (α2) and Asp99 (α3) and residues Asp70 (α1) and Arg102 (α3), which stabilize the relative orientations of these secondary structural elements, are indicated as dashed lines; (**b**) Cartoon representation of the solution structure of COA6 (PDB 6NL3) [52] (i) The structures are colored from blue at the N-terminus to red at the C-terminus. Cysteine residues are shown as yellow sticks; (ii) Residues Arg83 (α2) and Asp99 (α3) and residues Asp70 (α1) and Arg102 (α3) (labeled), which form salt bridges in the crystal structure, are located on ‘opposite’ sides of the molecule and separated by ~25 Å.

**Figure 3 ijms-21-06983-f003:**
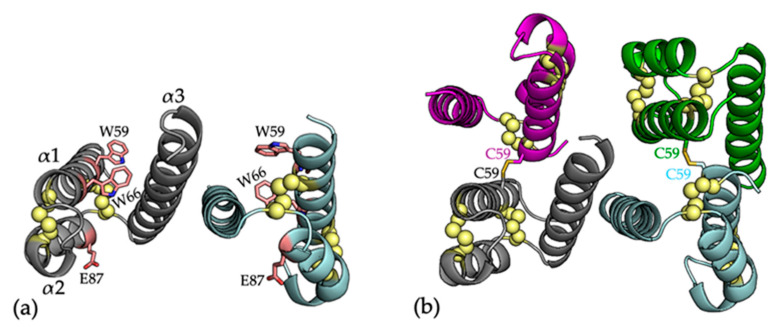
(**a**) The crystal structure of COA6, with residues that are mutated in patient samples shown as pink sticks (PDB 6PCE) [56]. Secondary structures are represented as cartoons with monomers colored in cyan and gray. Cysteine residues are shown as yellow spheres; (**b**) Cartoon representation of the crystal structure ^W59C^COA6 (PDB 6PCF) [56]. Individual protomers are colored in cyan, gray, purple, and green. Each monomer (cyan and gray) is linked to another monomer (green and purple, respectively) by an intermolecular disulfide bond (shown as yellow sticks) through the introduced Cys59 residue.

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
