# Peer review of "What Role Does COA6 Play in Cytochrome C Oxidase Biogenesis: A Metallochaperone or Thiol Oxidoreductase, or Both?"

_ijms, 2020, doi:10.3390/ijms21196983_

Round 1

Reviewer 1 Report

In the submitted review article the authors discuss the involvement of COA6 protein in the biogenesis of cytochrome c oxidase, a terminal oxidase of the mitochondrial respiratory chain.
Based on all the reviewed experimental evidence, they conclude that the exact role of COA6 is still questionable.

Given the authors expertise in the field of OXPHOS and COX, they discussed all aspects appropriately.
It should be perhaps noted in the text, that the alteration of copper homeostasis dramatically changes the iron-sulphur cluster formation thus has a broad effect on the mitochondrial function.

Simple scheme illustrating the oxidoreductase and/or copper binding of COA6 would serve as a nice graphical abstract. Space permitting, an overall COX assembly scheme hypothesis would put the role of COA6 into the perspective of the whole pathway.

Author Response

We thank this reviewer for the positive review of our manuscript. Minor points raised are addressed below:

1. It should be perhaps noted in the text, that the alteration of copper homeostasis dramatically changes the iron-sulphur cluster formation thus has a broad effect on the mitochondrial function.

We have added a statement (lines 135-136) and reference 35 to address this point. 

2. Simple scheme illustrating the oxidoreductase and/or copper binding of COA6 would serve as a nice graphical abstract.

To our knowledge, articles in IJMS do not include graphical abstracts. We would be happy to provide one if requested by the Editor.

3. Space permitting, an overall COX assembly scheme hypothesis would put the role of COA6 into the perspective of the whole pathway.

There are quite a number of such schemes already published. We feel ours would not be sufficiently distinct from those already published to warrant inclusion.

Reviewer 2 Report

This manuscript is a review article about how the assembly factor COA6 participates in COX biogenesis and plays a role in the biogenesis of the dinuclear CuA site in the COX2 subunit. Copper is essential for COX activity, but the molecular mechanisms of copper insertion to COX1 and COX2 subunits remain mostly unknown. COA6 is localized intermembrane space and contains CX9C–CX10C motif, indicating a redox active property, which could be a thiol oxidoreductase property or a copper (I) binding capacity. The manuscript is well written, and easy to follow.  It covers recent detailed information about the function of COA6 and it is informative to the audience in the field. I have only a few comments.

  1. In Figure 1c, the authors showed and described “Copper ion are shown as orange spheres. A cluster of two copper atoms, bridged by Cys196 and Cys200 residues constitutes the CuA site.” However, in the text on page 3, line 102, “The two copper atoms are bridged by two cysteine residues (Cys200 and Cys204). This should be Cys196 and Cys200?

Then, at line 104, “while the other copper atom is coordinated by residue His204 and the carbonyl group of Glu198 (Fig. 1c).

So, the reviewer is confused. Your figure shows His204. Probably it is better to use consistent residue number either from human or yeast proteins. Please fix this. Cys204 appears again at line 109 and also on page line 145.

  1. On page 4 line 146, what is AMSgel shift assay? You meant Electrophoretic mobility shift assay?

Author Response

We thank the reviewer for their positive comments. Specific points are addressed below:

  1. In Figure 1c, the authors showed and described “Copper ion are shown as orange spheres. A cluster of two copper atoms, bridged by Cys196 and Cys200 residues constitutes the CuA site.” However, in the text on page 3, line 102, “The two copper atoms are bridged by two cysteine residues (Cys200 and Cys204). This should be Cys196 and Cys200?

Then, at line 104, “while the other copper atom is coordinated by residue His204 and the carbonyl group of Glu198 (Fig. 1c).

So, the reviewer is confused. Your figure shows His204. Probably it is better to use consistent residue number either from human or yeast proteins. Please fix this. Cys204 appears again at line 109 and also on page line 145.

This has been corrected.

  1. On page 4 line 146, what is AMSgel shift assay? You meant Electrophoretic mobility shift assay?

We have added the full term for AMS (4- acetoamido-4′-maleimidylstilbene-2,2′-disulfonic acid) and corrected the terminology for the assay.